# The African Pastor as a Public Figure in Response to Gender-Based Violence in South Africa: A Public Pastoral Intervention

**Patrick Nanthambwe \*** and **Vhumani Magezi \***

The Unit for Reformational Theology and the Development of the South African Society, Faculty of Theology, North-West University, Potchefstroom 2531, South Africa
\* Correspondence: patricknanthambwe@yahoo.com (P.N.); vhumani.magezi@nwu.ac.za (V.M.)

**Abstract:** The burgeoning field of public theology has garnered significant scholarly attention. Amidst its multifaceted discussions, a recurring theme asserts that theology plays a vital and irreplaceable role in public discourse. This perspective contends that engaging with matters of public concern from a theological standpoint not only contributes meaningfully to public discourse but also shapes our understanding of the world, human existence, and the divine. Within the African context, particularly in South Africa, gender-based violence (GBV) remains a pressing societal issue despite government and organizational efforts. This article delves into the potential role of pastors as public figures in addressing the persistent challenge of GBV. It explores the implications of pastors assuming public roles within an African context and how this engagement can be instrumental in combating GBV. By drawing on literature related to public practical theology, pastoral care, and GBV in South Africa, the article advocates for proactive public interventions by pastoral ministries. Through synthesizing insights from existing scholarship, it contributes to ongoing discussions at the intersection of theology, pastoral practice, and societal issues, with a specific focus on addressing GBV in the unique South African context.

**Keywords:** African pastor; public; gender-based violence (GBV); pastoral intervention; pastoral care; South Africa

## 1. Introduction

The burgeoning field of public theology has become a focal point in scholarly discourse, capturing the attention of various scholars (Nzewi and Groenewald 2013; Forster 2015; Kim 2020). Amidst the multifaceted debates within practical theology, there is a consensus that this discipline should grapple with the complexities of faith in the contemporary world, seeking to provide meaningful insights into the myriad of challenges individuals face, spanning the realms of politics, society, religion, culture, health, and spirituality (Graham 2008, p. 13; Magezi 2019b, p. 119).

However, within this consensus lies a contentious debate regarding the public significance of religion. While many practical theology theologians endorse the idea that theology should actively address and offer tangible solutions to the diverse issues encountered in daily life, some scholars take a critical stance on the notion of religion playing a prominent role in the public sphere. Notable among them is Jens Köhrsen (2012), whose article titled "How religious is the public sphere? A critical stance on the debate about public and post-secularity" challenges the prevailing ideas about the impact of theology on public affairs.

In his critique, Köhrsen (2012) specifically targets the concepts of post-secularity, public religion, and desecularization. These notions propose that religion is experiencing a resurgence, gaining a more significant presence and influence in the public domain. However, Köhrsen (2012) argues that these concepts rely on an overly broad definition of

religion, allowing for the identification of religious elements in phenomena which may not inherently possess religious characteristics.

While his argument is based on the context of Western Europe, the same critical assessment needs to be applied in Africa and, in our case, South Africa, alongside asking earnest questions like how is theology impacting South African societies? We would like to admit that responding to such a subjective question is not easy considering the situation in South African public spaces. The rampant surge in crime, poverty, unemployment, corruption, gender-based violence (GBV), and many other vices in the country speaks volumes regarding the extent to which theology is being effective in South African communities (Magezi 2019b; Nanthambwe 2022; Resane 2019). Resane (2019, p. 2) observes the following:

> The prophetic voice in South African public space is deafeningly silent. Public theology seems to be conspicuous by its silence in the public arena of South African human affairs. Secularism, which is marked by indifference to, or rejection or exclusion of, religion and religious considerations, seems to have captured the minds.

The theology dilemma in South Africa mentioned above by Resane (2019) is also manifested in the way GBV is addressed within South African public spaces. In contrast to more secular African countries such as Mozambique and Ethiopia (Michael et al. 2024), South Africa continues to experience a significant prevalence of gender-based violence (GBV) (Mpako and Ndoma 2023; South African Government 2023). Despite being characterized as a religious nation, with Christianity as the predominant faith (Landman and Mudimeli 2022), what accounts for the persistent increase in GBV rates in South Africa?

This article delves into the pervasive issue of GBV in South Africa. It specifically examines the potential of pastors as public figures to contribute to resolving this entrenched problem. Focusing on the role of a pastor as a public figure within the African context, the article investigates how such a position can be harnessed to effectively combat GBV. It contends that the public engagement of pastors can serve as a vital intervention, addressing not only GBV but also various challenges afflicting South African communities. By elucidating the responsibilities and opportunities inherent to the pastoral role as a public figure, this article advocates for a robust public intervention by pastoral ministry.

This article assumes that public theology emphasizes the conversation to influence society. However, public practical theology and, to be specific, public pastoral care go beyond debates and conversations to develop practical intervention ministries where the publics are leveraged as spaces for ministerial interventions and constructive changes (Magezi and Nanthambwe 2022). This article takes the GBV issue as an element that can be an agenda for public conversation and debate to effect change in South African communities.

To underpin its argument, this article draws on the existing literature concerning public pastoral care, public practical theology, and GBV in the South African context. Through a synthesis of insights from these sources, this article aims to construct a compelling case for the necessity of public pastoral intervention in tackling the multifaceted challenges associated with GBV. This concise exploration contributes to the ongoing discourse on the intersection of theology, pastoral ministry, and societal issues, offering specific insights into addressing GBV in the unique context of South Africa.

## 2. The Problematization of the Issue of GBV in South Africa

GBV is one of the many challenges that South African communities are plagued with (South African Government 2022). Both traditional media and online social media platforms extensively deliberate on the pervasive issue of GBV perpetrated against women in South Africa (Oparinde and Matsha 2021, p. 1). Women and girls consistently withstand the worst of GBV occurrences (ibid.). For example, Munusamy (2013) narrates about 17-year-old Anene Booysen who was brutally attacked, raped, and disemboweled in Bredasdorp. In 2017, 22-year-old Karabo Mokoena went missing, and her body was later found burnt in an open field in Johannesburg (Saba 2017). Fast forward to 2021, 23-year-old law student

Nosicelo Mtebeni was brutally killed and dismembered, her remains concealed within a suitcase (Dayimani 2021).

These harrowing incidents reverberated throughout the nation, yet they represent only a fraction of the broader pattern. Recently, during the second quarter of 2023/2024, Police Minister Bheki Cele disclosed alarming crime statistics for South Africa (South African Government 2023). Within three months, the country recorded 10,516 cases of rape, 1514 attempted murders, and 14,401 assaults against female victims (ibid.). Tragically, 881 women lost their lives during the same period. These figures underscore the urgent need for comprehensive interventions to address violence against women in South Africa (South African Government 2023; Mpako and Ndoma 2023). Utilizing diverse platforms such as media outlets, human rights activists, organizations, and political leaders vocally condemn such acts of brutality. The civil society of South Africa has demonstrated considerable activism by orchestrating marches and rallies, emphatically addressing the pressing matter of violence against women within the nation (Oparinde and Matsha 2021, p. 1; South African Government 2022).

Various religious traditions have also played a role in implementing practical measures to combat GBV within the country (Resane 2024). Particularly noteworthy is the Christian faith, with churches utilizing pastoral care ministries to address GBV (Nanthambwe 2022; Nanthambwe and Magezi 2022; Resane 2024). For instance, pastors have become increasingly involved in activities such as raising awareness and providing education, offering counseling and support services, and engaging in theological discussions regarding the role of faith in addressing societal issues like GBV (Nanthambwe and Magezi 2022).

Despite South Africa being hailed for possessing one of the most progressive constitutions globally, the surge in GBV presents a stark contradiction to this notion (Sanni and Ofana 2021, p. 387). The year 2019 witnessed a notable increase in GBV incidents, prompting women and girls to organize marches and demonstrations in Johannesburg and Cape Town as a response (Nanthambwe 2022, p. 84; Oparinde and Matsha 2021). While 2019 was initially perceived as a particularly distressing year for GBV in South Africa, 2020, marked by the COVID-19 lockdown, emerged as yet another harrowing period, earning the grim designation of "another pandemic" for GBV within the nation (South African Government 2022). The lockdown period witnessed a distressing escalation in GBV cases, particularly accentuated in the rural regions of South Africa. What factors contribute to the pervasive nature of gender-based violence (GBV) in South Africa?

While multiple factors contribute to the prevalence of GBV, scholars such as Wa Muiu (2008), Gqola (2015), and Resane (2024) assert that its persistence in South Africa is deeply rooted in the cultural framework of patriarchy. Wa Muiu's (2008) research underscores how patriarchal norms and power structures perpetuate GBV by reinforcing unequal gender relations and legitimizing violence against women. Similarly, explaining the features of patriarchy, Resane (2024, p. 2) elucidates the following:

> The main features of patriarchy include domination, male supremacy, man's rule, control, man's advantage over women, and in some cases, negative attitudes towards females whereby females become tools to satisfy male ego. In patriarchy, men expect to be served by women to quench their desirable needs such as food, sex, and respect.

Resane (2024) provides a comprehensive overview of the key features of patriarchy, capturing its inherent power imbalances, systemic inequalities, and gendered expectations.

Gqola (2015, p. 2) highlights the stark reality faced by black women in South Africa, stating that "One out of nine black women living in South Africa have either been raped or are left to deal with the trauma of sexual abuse in silence." This grim statistic underscores the urgent need to address the systemic issues of gender inequality and patriarchal attitudes that fuel GBV in South African society.

Wa Muiu (2008, p. 83) posits that the culture of patriarchy in South Africa can be directly attributed to the legacy of apartheid, wherein the advancement of women's rights was intricately linked with the liberation struggle. Apartheid, as observed in South Africa,

was underpinned by hierarchical power structures based on class, race, and gender dynamics (ibid.). The concept of gender encompasses the socially constructed distinctions between male and female roles within society. Within the apartheid narrative, the archetype of the ideal white male was depicted as embodying traits associated with athleticism, outdoor pursuits, military service, and educational attainment (ibid.). Conversely, African men were esteemed for their intellect, education, and perceived belonging to the middle class (ibid.). White women were often objectified within society and the media, while African women were commodified and relegated to the role of progenitors of society, available for sexual exploitation by both white and black men (ibid.). Despite the legal framework of racial segregation and apartheid, white men maintained sexual dominance over women of both racial groups (ibid.). African women were relegated to servitude and discouraged from aspiring beyond their designated social status. Notably, sentences for theft were typically more severe than those for rape, particularly when the victim was not white, reflecting societal biases (ibid.). The normalization of male violence through the adage "boys will be boys" perpetuated a culture where violence among men was tacitly accepted as natural (Wa Muiu 2008, p. 8).

While there exists debate surrounding the extent to which apartheid contributed to GBV as opposed to the contention that GBV was inherent in African culture due to its patriarchal orientation, the undeniable reality is that GBV has emerged as a significant concern within the country, prompting widespread public apprehension (Nanthambwe 2022).

## 3. How Pastors Are Perceived in Africa

In many African communities, the role and persona of a pastor carry significant weight as they are perceived as prominent public figures (Baloyi 2024). Baloyi (2024, p. 1) astutely observes that, within the rich diversity of South African contexts, pastors are universally recognized as models of virtue, actively contributing to the fabric of their communities through their embodiment of positive, ethical, and moral principles. This elevated status inherently positions pastors as conspicuous public figures within the African setting, their influence extending far beyond the walls of their places of worship (ibid.). As public figures, pastors in Africa are considered models, giving examples to society on how to live a life.

Chivasa (2017) highlighted this sentiment, asserting that the pastoral vocation in Africa is intricately intertwined with public visibility and societal prominence. Indeed, Chivasa (2017, p. 3) contends that it is inconceivable for an individual assuming the mantle of a pastor not to become a public figure, given the deeply ingrained cultural expectations and societal reverence bestowed upon religious leaders. Furthermore, Magezi (2019b, p. 5) corroborates this perspective, emphasizing the inherently public nature of the pastoral role in African communities. Magezi (2019b) underscores that the very essence of being a pastor in Africa precludes privacy and individualism, mandating active engagement with the public sphere. This assertion highlights the significance of pastors' active involvement in public life within African contexts. Pastors are not isolated individuals but play crucial roles in engaging with communities and addressing societal challenges. This agrees with what Vanhoozer and Strachan (2015, pp. 16–17) advised, in that the pastor should be a theologian by saying what God is saying in Christ, and, at the same time, they should publicly be involved in and for the community.

To understand the notion of public theology or the public role of the pastor one must ask the following question: what is public in public theology or the public ministerial of a pastor? Baloyi (2024, p. 1) aptly explains the following:

> Communities expect pastors to be helpful not only at funerals, but also during other disasters and unforeseen calamities. For example, on 23 September 2023, the Collins Chabane mayor convened the pastors in Malamulele Community Hall to pray for many things, including an end to crime and accidents on the roads.

Baloyi's (2024) statement above shows the multifaceted role of pastoral leaders, which include pastors as community supporters, and spiritual leaders who actively engage with and contribute to the well-being of their communities beyond the confines of the church

walls. It highlights the significant impact that pastors can have on societal values and the extent to which they are expected to be involved in addressing community needs and concerns.

In asserting the recognition of pastors as public figures in Africa, Nanthambwe and Magezi (2022) elucidate, in their article titled *Community development as an embodiment of pastoral care in Africa: A public practical theology perspective*, four essential pathways through which pastoral care effectively engages with the public domain within the African setting. These pathways encompass the following: (1) Public healing, where pastoral care extends beyond individual healing to address communal well-being. (2) Giving people a voice, where pastors empower individuals to express their needs and concerns, amplifying community voices. (3) Church-led social mobilization, where pastors and their congregations play a pivotal role in mobilizing communities for positive change. And, finally, (4) leadership and transformation, where pastoral leadership extends beyond the pulpit, impacting community transformation (Nanthambwe and Magezi 2022, pp. 11–15; Magezi 2020; Louw 1997). Louw (1997) summarizes these when he contends that the care provided by pastors in Africa must adopt a contextual and community-based approach that is focused on existential aspects.

In communal areas, pastoral work entails the pastor assuming a significant leadership role within the community. This assertion is substantiated by empirical evidence from studies conducted by Magezi (2019b), Nanthambwe and Magezi (2022), and Rodgers (2021). Notably, Rodgers' (2021) research in Kenya highlights the pivotal function of pastors as community leaders, particularly in facilitating dialogues and decision-making processes. The collective findings of these scholars converge to emphasize the profound impact of pastoral work in communal settings. As custodians of communal values and catalysts for change, pastors play a prominent role in guiding their communities toward a more cohesive and promising future (Magezi 2022, p. 15). This underscores the transformative potential of pastoral leadership within the African context.

Furthermore, pastors and churches in Africa are considered sources of information, hence making their ministry more public. As posited by Öhlmann et al. (2016), church leaders, namely, pastors, alongside their ecclesiastical communities in Africa, emerge as pivotal conduits of information, particularly for individuals devoid of conventional access channels. This encompasses disseminating crucial insights on medical treatment and services. Notably, even modest congregations or those lacking physical church infrastructure serve as significant platforms for disseminating knowledge on HIV and AIDS, including strategies for prevention and potential remedies (Öhlmann et al. 2016, p. 11).

The perception of pastors as public figures in Africa underscores their multifaceted roles as spiritual guides, community leaders, and sources of information. Their active engagement with public life and community affairs reflects the dynamic nature of pastoral ministry within African contexts, with significant implications for societal values and development.

## 4. Understanding the Notion of Public in the South African Context

The preceding section has provided a concise examination of the perception of pastors within the African cultural framework. It has been elucidated that pastors in Africa are perceived as prominent figures within society. This section prompts consideration regarding the comprehension of public spaces within the South African context. Exploring the concept of the public is crucial for grasping the concept of public theology, thus facilitating a clearer understanding of the pastor's public role, which can be used to address GBV in South Africa. Our inquiry commences with an examination of the concept of the public.

### 4.1. What Are Publics?

According to Day and Kim (2017, p. 11), understanding how theology engages with the publics commences with the definition of public. Day and Kim (2017, p. 11) argued the following:

The actual process of theological engagement with public issues has begun with defining 'public'. As long as public is perceived as the public—amorphous and monolithic—any attempt at theological engagement will be abstract and irrelevant. The premise of public theology is that the discourse does not remain within a rarefied community of academic theologians, which would only be self-serving.

Day and Kim's (2017) statement underscores the importance of defining what is meant by "publics" before one begins engaging theology with public issues. Habermas (2006, p. 73) gives the following definition of what is meant by public sphere:

By "the public sphere" we mean first of all a realm of our social life in which something approaching public opinion can be formed. Access is guaranteed to all citizens. A portion of the public sphere comes into being in every conversation in which private individuals assemble to form a public body.

According to Day and Kim (2017, p. 2), publics represent social arenas conducive to dialogue, wherein cohesion arises amidst and, indeed, due to the diversity and occasional conflicts they encompass. They are likened to forums or agoras in ancient Greece, facilitating encounters with diverse perspectives and fostering engagement with the unfamiliar.

Morton (2004) elucidates the distinction between publics and communities, highlighting the importance of differentiation. In community settings, the emphasis rests on shared characteristics and commonalities among individuals, whereas, within publics, the focus shifts towards recognizing and embracing diversity and differences. Day and Kim (2017, p. 12) expound upon this concept by delineating publics as distinct from communities, emphasizing that publics may share linguistic traits but primarily serve as arenas for social dialogue and interaction. These spaces coalesce not in spite of but rather due to the inherent disparities and conflicts they encompass (ibid). Day and Kim (2017) further posit that publics are characterized by a dynamic interplay of questioning, doubting, and challenging, alongside affirming, confirming, and agreeing (Magezi 2022).

Day and Kim (2017, p. 12) present the following publics, identified by different public theologians:

- Tracy (1981) identified three publics, namely, the academy, the wider society, and the church.
- Stackhouse (1997) identified academic, economic, religious, and political sectors.
- Benne (1995) added law as another public to Tracy's and Stackhouse's publics.
- Dirkie Smit (2003) identified four publics—namely, political, economic, civil society, and public opinion.

Magezi (2022, p. 10) observed that the aforementioned publics are evidently shaped by individual contexts, emphasizing the fluid nature of these social arenas. They are not static entities but rather dynamic spaces that evolve over time. For instance, Dirkie Smit's conception of publics, rooted in his South African background, mirrors the dynamic nature of the South African context.

*4.2. South African Public Spaces*

From the discussion above, there are several publics that are identified by theologians. What makes the publics exist is the nature of engagement they present (Magezi 2022). For one to understand the public role of a pastor in Africa, one needs to understand the nature of the African publics. According to Magezi (2022), there are specific publics that are specific to Africa.

Before zeroing in on the specific publics in South Africa, a general understanding of the South African social outlook is particularly important. Nanthambwe's (2022) study shows that there are three categories of people living in South Africa. These are the underdeveloped, the developing, and the developed. His study (Nanthambwe 2022) shows that, mostly, the underdeveloped live in rural areas and townships, while the developing and the developed live in the suburbs of the country. In this setting, the visibility of pastoral

involvement in publics is minimal in suburbs compared to townships and rural areas (Magezi 2022; Nanthambwe 2022).

In addition to the publics presented above by Tracy, Stackhouse, Benne, and Smit, Magezi (2022) introduces an additional layer to this discussion by advocating for the recognition of African traditional forums as a distinct public. These forums, deeply rooted in African cultural heritage, play a conspicuous role within the country. They encompass customary laws, judicial systems, conflict resolution mechanisms, and property rights, all of which are distinct from formal state institutions. The importance of traditional forums in Africa is echoed by Mangisteab (2023), pointing out their relevancy and indispensability in several areas for the governance of Africa's traditional economic sector. Magezi (2022, p. 11) states the following:

> The word traditional does not suggest primitive but historical indigenous cultural patterns and ways of living whereupon communities were governed before Western democracy. These traditional forums have continued to exist albeit with moderation in some respects particularly the checks, balances and restraints brought by country laws. Within South Africa, traditional community forums are governed by Act No. 41 of 2003: Traditional Leadership and Governance Framework Amendment Act, 2003. Among other things, the Act provides "for the recognition of traditional communities; to provide for the establishment and recognition of traditional councils. . ." The African traditional forums as a social public are not isolated from the other publics but integrated. Issues discussed at the forums are often subjects of conversations in other publics such as civil society, like gender-based violence.

This quotation elucidates the role of African traditional forums in the preservation and guardianship of culture. Minister Dlamini Zuma (2021) attests to the significance of these forums and the participation of traditional leaders in addressing issues that adversely affect individuals' lives. Zuma (2021) assertively argues that traditional leaders serve as custodians of culture, customs, and tradition, while also fulfilling a pivotal role in educating their communities about societal challenges, particularly raising awareness about gender-based violence. Additionally, they actively advocate for the promotion of human rights among vulnerable populations (Zuma 2021).

The question is the following: how do these forums manage to achieve the above feats? According to Magezi (2022, p. 13), these forums engage in discussions and debates, among other activities, to address local community issues. These issues encompass disputes, community transgressions, crimes, and pressing challenges. Magezi (2022) pointed out that these forums exhibit several distinctive features including the following: (1) Plurality of religious practices: They embrace a diversity of religious expressions. (2) Awareness building and updates: They disseminate information about ongoing events. (3) Gatekeeping of social values: They safeguard and preserve traditional cultural norms. (4) Assertion of social values: They actively promote and reinforce communal values. (5) Setting community norms: They contribute to defining what is acceptable or unacceptable behavior. And, finally, (6) political influence and conflict: They serve as arenas for political persuasion and may witness tensions between democratic institutions and customary laws.

However, despite their positive aspects, these spaces can also suppress certain voices. Chiefs often wield influence within traditional forums, and contradictions arise when national laws clash with cultural traditions. Moreover, these forums serve as negotiation grounds, where community members increasingly assert their constitutional rights vis à vis traditional structures. In sum, these traditional forums harbor both toxic and life-affirming practices, reflecting their multifaceted nature within African contexts (Magezi 2022).

The other question that needs to be answered is the following: what role do traditional forums play in promoting human dignity and well-being? Mangisteab (2023) and Magezi (2022) elucidate that traditional forums represent a crucial social arena within African societies, notwithstanding their limitations. They are readily accessible and situated within the proximity of community members, facilitating the addressing of their challenges and

urgent public concerns. Being centered around community members, these forums are adept at identifying and responding to local needs effectively. More importantly, the forums provide people with a sense of respect and dignity, as they are grounded in people's cultures, customs, and norms (Magezi 2022; Mangisteab 2023).

It is in these spaces that this article argues for pastoral influence to combat GBV in South Africa. Given that pastors are widely recognized as public figures across many African nations (Baloyi 2024; Chivasa 2017; Magezi 2019a; Vanhoozer and Strachan 2015), they wield a noteworthy influence over public affairs. However, how do pastors exert their influence in these settings? To answer this question, we must first explore the concept of public theology, which serves as a vehicle through which pastors can effectively engage with public issues.

*4.3. What Is Public Theology?*

According to Dreyer (2004, p. 919), public theology is defined as follows:

> A fairly recent term referring to a theology which critically reflects on both the Christian tradition as well as social and political issues. This dialogue is seen to benefit both theology and society.

Dreyer (2004) goes on to differentiate practical theology from public theology by asserting that not all practical theology qualifies as public theology, meaning that it is not necessarily directed towards a non-ecclesial general audience. This suggests that the intended audience for public theology is not limited to the "church" but extends to the broader "public". Mannion (2009, p. 122) contributes to clarifying the scope of public theology by stating the following:

> So, to chart briefly the scope and range of public theology, we can begin by saying that most contributors to such discourse would agree that public theology is theology that is social, political, and practical. But I would argue that at its best public theology involves theological hermeneutics in the service of moral, social, and political praxis.

Dreyer (2004) and Mannion (2009) outline that public theology focuses on issues related to the public, rather than exclusively on church matters. Kim (2017, p. 40) elaborates on this concept by highlighting that public theology emerges from theology's engagement with political and economic spheres, subsequently expanding to include civil societies and other domains of public life. Smit (2017, p. 75) adds that public theology distinguishes itself from church-centric theology by actively engaging with the public sphere. Unlike discussions confined to religious spaces or academic circles, public theology is dynamic and practiced in the wider world. It unfolds openly within society and communities, encompassing all aspects of life and inviting participation from everyone in the public domain. This form of theology extends beyond church internal dialogues to reach individuals in public spaces, from streets and markets to public forums.

But, to what extent does public theology engage with public issues? Kim (2017, p. 40) posited that public theology is a "critical, reflective and reasoned engagement of theology in society to bring the kingdom of God, which is for the sake of the poor and marginalised". Kim's (2017) assertion stresses that public theology is primarily concerned with those on the fringes of society. It seeks to rectify theology that is irrelevant, distant, and detached by grounding it in the daily realities faced by people (Magezi 2020).

In essence, public theology critically examines the role of theology in shaping societal dynamics (Magezi 2020). It is because of problems like poverty, GBV, racism, oppression, exploitation, corruption, and many other vices in society that public theology becomes relevant.

**5. How Can Pastors in South Africa Utilize Public Theology to Address GBV in Public Spaces? The Implications of Pastoral Intervention against GBV**

There is a need for intervention at multiple levels to deal with GBV in South Africa. The questions to ask are the following: What role can pastoral care play in this situation? How can a pastor enter a public fray and make a contribution. What is this contribution?

Firstly, the potential for pastors to play a role in improving the quality of life for individuals, including tackling GBV, in South African public spheres notably within traditional forums is evident. Churches hold strategic positions owing to their proximity to communities (Magezi 2019a, 2022). This proximity allows pastors and other church leaders to be actively engaged from an informed standpoint. Magezi (2022, p. 14) highlights the following:

> The historically developed and written resources on theology and ethics assist pastors to influence and infuse positive values. In many communities, some churches have developed a track record of engaging community structures, which gives pastors credibility. Also, Christianity and theology have historically taken a liberatory stance to assist people to be free and pursue flourishing and human-hood pathways in life. Different theologies such as Liberation, Feminist, Womanist, Black and White have emerged to address social experiences of people. Furthermore, faith-based groups such as South Africa Council of Churches (SACC) have also been actively advocating for social good. By its very nature, theology and churches discourage lawlessness and violence. This foundational principle encourages people to uphold law and order, spread and uphold positive message and life-giving values. However, while church and theological message encourages law and order, in situations of evil and community ill, it also carries a message of confrontation and healing. Thus, there is co-existence and a dual message of confrontation and peace.

Christianity offers a message that opens up possibilities for enhancing the quality of life through meaningful and wise engagement within traditional forums. For instance, in the context of South Africa's pervasive GBV, pastors, as public figures, can engage with custodians of traditional culture through non-confrontational, life-affirming conversations aimed at transforming perspectives and perceptions (Setswe et al. 2009). By participating in public discussions, advocating for policy changes, and promoting awareness, pastors can contribute to societal transformation.

Secondly, they can achieve this goal by advocating the message of humanness in all South African public spaces. The perspectives within public spheres vary significantly. In the African context, one prevalent belief among many African men, including those in South Africa, is the inferiority of women to men (Coetzee 2001; Hanyane and Ahiante 2022; Jansen van Rensburg 2021). Unfortunately, this perception is widespread in numerous African traditional settings and contributes significantly to GBV, as women are often viewed as objects for fulfilling men's sexual desires, resulting in their violation. Pastors must utilize public spaces as platforms to disseminate the biblical message that all human beings possess inherent dignity by being created in the image of God (Gen 1:26–28), thereby advocating for the equal treatment of women and men (Agang 2020).

Thirdly, to embody the belief that all individuals are created in the likeness of God, pastors should encourage their congregations to establish ministries that foster coexistence between men and women, while also promoting and nurturing mutual respect (Nel 2019). Pastors in South Africa should strive to transform faith communities into spaces where women are empowered. It becomes challenging for the broader public to embrace the concept of equality if churches themselves fail to implement it (Nel 2019; Plaatjies-Van Huffel 2011). Therefore, the effectiveness of pastors' advocacy for equality hinges on the visible implementation of such principles within churches, where women should be empowered to attain equal status with men. Regrettably, this goal has not been achieved, as expressed by Nel (2019) and Chisale (2020). Although they do not provide specific percentages regarding the proportion of women in ministry in South Africa, Nel (2019) and



Chisale (2020) have criticized the predominance of men in leadership positions within many Evangelical churches. However, female pastors in the country have become increasingly conscious and engaged in efforts to address GBV by providing counseling and support to victims, in comparison to their male counterparts (Wagner-Ferreira 2011).

Fourthly, pastors are tasked with advocating for change by identifying and confronting sinful behaviors. The following question then arises: how should pastors go about this responsibility? Walker (2019) highlighted that, among the various roles of public theology, one significant aspect is its duty to critique the sinful aspects of the democratic state in public discourse. Pastors must fearlessly denounce societal injustices, particularly the persistent mistreatment of women and children. Walker (2019, p. 39) aptly suggests that public theology serves as a platform for articulating societal ideals, correcting societal wrongs, and offering hope amidst deteriorating societal structures. Pastors can instill hope by providing essential emotional support, counseling, and guidance to survivors of GBV and their communities. Unfortunately, there is silence on many church platforms regarding GBV. Nanthambwe's (2022) study shows that many participants in the study, albeit church leaders, did not acknowledge GBV as a major challenge in their communities. What is an explanation for this? The failure of participants to acknowledge gender-based violence (GBV) raises significant concerns about how church leaders perceive this issue. Are church leaders even recognizing GBV as a problem? Peter Manzanga's (2020) study within the Zimbabwean church context revealed that, while GBV poses a serious challenge in many African communities, it remains a topic often avoided in churches. In interviews conducted for Manzanga's (2020) study, participants expressed surprise at a pastor (Manzanga) openly addressing GBV with women. The study in question observed that churches tend to maintain a culture of silence regarding GBV, evident in the lack of teachings from the pulpit on the subject in many African churches. It is rare to hear a sermon on GBV delivered on a Sunday. But why do African pastors, including those in South Africa, not publicly condemn GBV? Shingange (2023, p. 4) suggests that one reason is their adherence to a distorted theology that aligns with cultural norms and values. Within many African cultures, women are often regarded as inferior (Hanyane and Ahiante 2022; Jansen van Rensburg 2021), and crimes against them are frequently not taken seriously in public spheres, leading pastors to hesitate in addressing GBV publicly. By remaining silent on the issue from the pulpit, pastors inadvertently perpetuate the values and worldviews that contribute to GBV within the country. This must change, and GBV and other vices need to be confronted on the pulpit.

Fifthly, pastors and their congregations should strive to collaborate with other societal institutions in the fight against GBV in the country (Nanthambwe 2022). GBV is a pervasive issue nationwide, and it would be erroneous to believe that churches and their pastors alone can effectively address it (Magezi and Manzanga 2019; Manzanga 2020; Nanthambwe 2022). Nanthambwe's (2022) study conducted in Johannesburg indicates that many individuals recognize the significance of churches partnering with non-governmental organizations (NGOs) to tackle various community challenges, including GBV.

When religious institutions like churches join forces with other groups, including secular organizations, to tackle shared concerns, they experience numerous advantages (Magezi 2017). Throughout the Bible, God achieves His objectives through diverse means, including individuals, armies, and unconventional allies like prostitutes and borrowed donkeys, and the church reflects this diversity (Fields 2014, p. 2). According to Magezi (2017, p. 11), churches must devise additional strategies to tap into external resources from established networks to complement community efforts. It is crucial for churches to cultivate partnerships with the government and secular development organizations to address the challenges afflicting local communities (Nanthambwe 2022, p. 260).

By partnering with various societal institutions, including traditional platforms, to fight GBV in South Africa, pastors, and their congregations stand a greater chance of succeeding in their endeavors to address GBV in the country.

## 6. Conclusions

There is enough evidence to support that theology has a significant impact in public spaces. The question that this article endeavored to answer is how can pastors as public figures contribute to resolving the entrenched problem of GBV in South Africa. In answering this question, this article presented GBV as a huge challenge affecting communities in Africa. Furthermore, how people view pastors in the African context was discussed. This article showed that pastors in Africa are viewed as public figures and have a tremendous impact in influencing their societies.

The publics, especially the African traditional forum, and their potential as spaces where public issues can be resolved were discussed in this article. A brief conceptualization of public theology was presented, which helped us to come up with ways in which pastors in South Africa, utilizing public theology, can help address the issue of GBV. This article has argued that, while theology has public influence, pastors in South Africa must utilize the opportunities available and combat GBV in the country.

**Author Contributions:** Conceptualization, P.N. and V.M.; methodology, P.N. and V.M.; resources, P.N.; writing—original draft, P.N. and V.M.; writing—review and editing, P.N. All authors have read and agreed to the published version of the manuscript.

**Funding:** This research received no external funding.

**Data Availability Statement:** No new data were created or analyzed in this study. Data sharing is not applicable to this article.

**Conflicts of Interest:** The authors declare no conflicts of interest.

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
