# Peer review of "The African Pastor as a Public Figure in Response to Gender-Based Violence in South Africa: A Public Pastoral Intervention"

_religions, doi:10.3390/rel15050609_

Round 1

Reviewer 1 Report

Comments and Suggestions for Authors

This is a very interesting paper addressing an important social issues and how religion can help to solve it. The argument is theoretically solid and justified with proper bibliographical references.

1. The main question is wether a pastor can be socially relevant in preventing and addressing gender based violence in South Africa, and the thesis is "yes, a pastor can be relevant". The argument is solid and well referenced. 2. A gap could be a comparison with more secularized societies, but it is not properly requested. 3. Regarding methodology, providing precise data about gender based violence episodes and pastor public interventions would be effectively welcomed.

Reviewer 2 Report

Comments and Suggestions for Authors

In theoretical and methodological terms, the article is well founded and presents a good problematization of the object with updated data.

It discusses the notions of public, public space and public theology and discusses how South African society views the public figure of the pastor.

It proposes paths for public action by pastors regarding GBV and compares the situation with other countries.

Some questions:

Why don't South African pastors act more publicly against GBV?

What is the percentage of female pastors and what is their decisive role (or not) in this issue of GBV?

To what extent do pastors reinforce values ​​and worldviews that fuel GBV?
